# Multitopic Coherence Extraction for Global Entity Linking

**Chao Zhang** [1,2,*] **, Zhao Li** [1,2,3] **, Shiwei Wu** [4] **, Tong Chen** [4] **and Xiuhao Zhao** [1,2]

1   School of Computer Science and Technology, Qilu University of Technology (Shandong Academy of Sciences), Jinan 250353, China
2   Shandong Computer Science Center (National Supercomputer Center in Jinan), Jinan 250013, China
3   Shandong Key Laboratory of Computer Networks, Jinan 250014, China
4   Evay Info, Jinan 250101, China
*   Correspondence: accumuchao@gmail.com

**Abstract:** Entity linking is a process of linking mentions in a document with entities in a knowledge base. Collective entity disambiguation refers to mapping of multiple mentions in a document with their corresponding entities in a knowledge base. Most previous research has been based on the assumption that all mentions in the same document represent the same topic. However, mentions usually correspond to different topics. In this article, we proposes a new global model to explore the extraction of multitopic coherence in the same document. Herein, we present mention association graphs and candidate entity association graphs to obtain multitopic coherence features of the same document using graph neural networks (GNNs). In particular, we propose a variant GNN for our model and a particular graph readout function. We conducted extensive experiments on several datasets to demonstrate the effectiveness to the proposed model.

**Keywords:** entity linking; graph neural network; graph attention network

## 1. Introduction

The fundamental aspect of the entity-linking task is to correctly link entity mentions extracted from a document to the corresponding entity objects in the knowledge base. Entity linking is an essential part of natural language processing. It is a prerequisite step in many natural language processing tasks, such as question answering [1], relation extraction [2], and knowledge base-based question-and-answer systems [3].

Most current entity-linking tasks use a combination of two-stage models, i.e., local and global models [4–8]. The local model models the context of each mention in the document. In contrast, the global model considers all mentions in the entire document and models the global coherence of topics across all mentions [9]. During the modeling process, the similarity of entity mentions and candidate entities is calculated in terms of multiple features. In current, deep-learning-based entity-linking research, the most used features for ranking candidate entities are prior popularity, context similarity, and topic consistency. Prior popularity refers to the probability that a mention is linked to a candidate entity without considering to the context of the mention; the use of popularity alone can achieve an accuracy of more than 70% [5]. With the context similarity method, in order to obtain more hidden information about the current mentions and calculate the similarity with the candidate entity descriptions, the entity mentions and the corresponding contexts are encoded using a neural network to obtain the embedding of the current entity mention contexts. Under the topic consistency method, the two abovementioned features mentioned are the local similarity between the entity mention and one of the corresponding candidate entities. Each entity mention in the document can be independently connected to the corresponding candidate entity according to the two features mentioned above. These features are also called local features. According to Cucerzan, an increasing number of entities can be connected [10], with a focus on the global topic consistency among the

mentions of the document ensemble and the uniform association of these entity mentions. The global topic consistency feature is based on the assumption that mentions appearing simultaneously in the same document are topologically consistent entities and model the global consistency of topics in all mentions [9]. Most previous work tasks assumed that the topics in the same document are coherent and that there is one and only one topic [11,12]. Here, we consider the possibility of two or more topics appearing in the same document at the same time. That is, there may be two or more inconsistent topological structures in the same document. Entity mentions in a document can belong to different topological structures, and the embedded features extracted by the neural network for each topological structure may be inconsistent. For example, in the sentence, "Apple's Steve Jobs likes to eat apples made by Dole." Contains two topics. One is the topic of Apple, the company. The first mention of Apple and the mention of Steve Jobs belong to this topic. The second is the topic that represents apple (the fruit) and the mention (Dole); and the second mention of apple falls under this topic. However, the current global topic consistency model only considers globally unique topics that may affect the linking of entities. If only globally unique topics are considered, the model will likely to connect both apple mentions to the candidate entity Apple based on globally unique features. Such errors accumulate and lead to suboptimal results in the final global linking.

To address the shortcomings of existing global models, we propose a multitopic coherence extraction model to account for a high probability of sharing similar or identical topics between mentions in documents. The proposed global approach is designed to integrate the features between all candidate entity pairs. The collective underlying information between these highly similar mentions can be used as critical information to obtain a coherent feature representation of the hidden topic. In this model, we consider both mentions, their contextual information, and entity information in the target knowledge base. The model is divided into two parts. First, we assume that there are multiple topics in some documents. The mentions are composed of different mention association graphs according to the similarity between the mentions in the same document. Each mention association graph is a topic in the current document. The features of this mention association graph are obtained using a GNN. The second part forms a candidate entity association graph with all the candidate entities of all mentions under the current mention association graph. For this graph, we use another GNN to capture the globally consistent features of the current topic.

In the first part, the entity mentions in a document are fully connected as the nodes of the graph. The embedding similarity between the mention pairs is calculated as the edge weight to form a fully connected entity mention association graph. The key aspect of the problem is how to divide the fully connected entity association graph into one or more topic subgraphs based on the topic consistency between mentions. Considering multiple global topics, if we choose to divide the topics in the text according to a fixed number, it is very likely that there will be too many or too few split topics, which will significantly affect the topic feature extraction. Therefore, we adopt the method of dynamically determining the number of topics. That is, by calculating the similarity of all entity mentions, using hyperparameters to compare with the similarity threshold, the entity mentions that exceed the threshold are divided into the same topic, allowing the model to learn to determine how many topics are contained in each document. Numerous experiment prove that our dynamic approach can effectively classify topics with coherent entity mentions into the same topic.

In the second part, after obtaining the feature of the mention association graph, we do not use the feature directly as the topic feature of the current topic. The feature is associated with all candidate entities under all entity mentions associated with the current topic to obtain the candidate entities association graph. Feature extraction is performed on the graph to obtain the accurate feature vector of the current topic. Finally, we take the obtained topic consistency feature as the main feature to participate in the final global model result calculation to obtain the ranking of candidate entity mentions for each entity.

Because we are only constructing the global model, we compare the baseline model with the same local model. A large number of experiments on multiple datasets demonstrate the validity of our model.

Our contributions are summarized as follows:

1.  We propose a new hypothesis that there may be more than one topic consistency topology in a document. We propose a multitopic global consistency extraction model. The model adopts a new method of constructing an entity mention association graph and candidate entity association graph.
2.  We adopt a new graph readout approach and a variant version of a GNN for our model to construct a mention association graph.
3.  Extensive experiments using our global model on a benchmark with the same local model show that our global model achieves competitive results.

## 2. Problem Definition

Assume a document (*D*) containing entity mentions set $M = \{m_1, m_2, \ldots, m_T\}$, where for each mention, $m_i (1 \leq i \leq T)$, the candidate entity is $\Gamma(m_i) = \left\{ e_i^1, e_i^2, \ldots, e_i^{N_j} \right\} (N_i \geq 1)$. The purpose of entity linking is to connect each mention of $m_i$ to its corresponding gold entity, $e_i^* \in \Gamma(m_i)$. Entity-linking tasks are usually divided into two steps. The first step generates the set of possible candidate entities, $\Gamma(m_i) = \left\{ e_i^1, e_i^2, \ldots, e_i^{N_i} \right\} (N_i \geq 1)$. The second step is to sort the candidate entities. The candidate entities are sorted based on the combination of the local ranking score and global ranking score. The local ranking score is obtained from the local model, and the global ranking score is obtained from the global model.

## 3. Related Work

In this section, we focus on local and GNN modeling methods. Our work focuses on the construction of a global model, and the local model uses the local model proposed by Ganea and Hofmann [5].

### 3.1. Local Model

The local model was proposed by Ganea and Hofmann [5]. An attention mechanism is used based on the assumption that only part of the keyword information in the mention context can be exploited for the representation of mentions. These words help to reduce the noise of mentions and improve the ambiguity of mentions. In this study, a keyword is defined as a word in the mention context that is closely related to at least one candidate entity of the given mention.

We assume a local context for mentioning, and for each of its candidate entities, the local ranking score is calculated as follows:

$$\Psi(m_i, e_i^{N_j}) = f(h_{e_i^{N_j}} B h_{c_i}) \tag{1}$$

where $h_{e_i^{N_j}}$ is the embedding vector of the candidate entity, *B* is the diagonal matrix, a trainable parameter, and $h_{c_i}$ represents the embedding mention in the local context. $f(\cdot)$ is a function for which the attention mechanism is used. Ganea and Hofmann believe that only a few words in the context of entity mentions are effective for interpreting vague information about entity mentions. The values in the diagonal matrix represent the weight values of different words. Through this model, we can obtain the similarity score of each entity mention ($m_i$) with each candidate entity ($e_i^{N_j}$). The candidate entity with the highest score is ultimately selected as the target entity. In this process, each entity mention and its candidate entities are calculated independently, and the feature information of other entity mentions and other candidate entities is not considered.

### 3.2. Global Model

According to Ganea and Hofmann [5], each candidate entity is fully connected to another candidate entity in pairs, a joint probability distribution is defined using conditional random fields, and the marginal probabilities are directly optimized using truncated fitted LBP propagation applied to a fixed number of iterative message passes.

Le and Titov [6] pointed out that a single bilinear form score is too simple to uncover the complex semantic association information between entities. Therefore, they proposed the addition of multiple relations for each entity the use of different forms to assign different weight values to the relations, on top of the same LBP is added to achieve more competitive results at a lower cost.

### 3.3. Graph Neural Networks

The concept of graph neural networks was originally proposed by Gori et al. [13]. The concept of a complete graph neural network was subsequently developed and by Massa et al. [14]. Early approaches to graph neural networks mainly used recurrent neural units to aggregate information from the collocated nodes and eventually generated a representation of the target node. The concept of a spectral convolutional neural network (GCN) was proposed by Defferrard et al. [15]. Later, message-passing neural networks generalized these GCN-based approaches, defining a new paradigm for graph neural networks consisting of two steps: message passing and graph readout. Later, researchers proposed the concept of a graph-gated neural network (GGNN) [16]; the biggest modification of this neural network is the use of gated neural units and fixed T-step cycles. In the aggregation phase of message passing, the gating unit is used to control which features of the leader nodes should be aggregated.

The current GGNN is suitable for the sequence graph model. In order to adapt to our task, we improve the GGNN, from the previous order of information transfer to only consider the information of the node in degree. Because our entity mentions each node in the association graph, each node is bidirectionally associated, which cancels the sequence information. The information of each node can also be transmitted to the neighbor node for information aggregation.

The current graph readout function is a simple linear addition of the features of all nodes, resulting in the loss of a considerable amount of original feature information of the nodes; therefore, it is not conducive to capturing the entire graph feature. We propose a new graph readout method, which will be described in detail in the following chapters.

Cao et al. [17] applied a graph convolutional network (GCN) to the entity-linking task, which proved effective. Fang et al. [18] used a graph attention network to capture global topic coherence. The graph attention network can dynamically obtain important information about different neighbor nodes through a special self-attention mechanism. The existence of the graph structure can combine the same topic. All candidate entities are associated with each other, so the correlation information between candidate entities can be better captured.

Junshuang et al. [12] proposed a dynamic GCN model, arguing that entity mention connectivity graphs should consider the contextual information of entity mentions and the structural information of entity mention relatedness. We propose a dynamic GCN paradigm applied to entity connectivity, whereby the graph structure input into the GCN is dynamically computed and modified during the model training process.

## 4. Multitopic Coherence Model

The overall framework of our model consists of two main parts: the mention association graph and the candidate entity association graph. The mention association graph is generated from the mention full connectivity graph and represents the classification relationship of the topic. Different mention association graphs can be used to extract the potential semantic features of the mention collected by our variant GNN. Furthermore, this semantic feature is used as the association feature of the current topic. The candidate

entity association graph comprises all the candidate entities under the current topic and the obtained mention entity association graph features. Then, the global coherence features of the current topic are obtained through the graph attention network (GAT). Finally, the global score of each candidate entity under the current mention set is calculated based on the similarity between the coherence feature of the current topic and the representation vector of each candidate entity. The overall architecture of our model is illustrated in Figure 1.

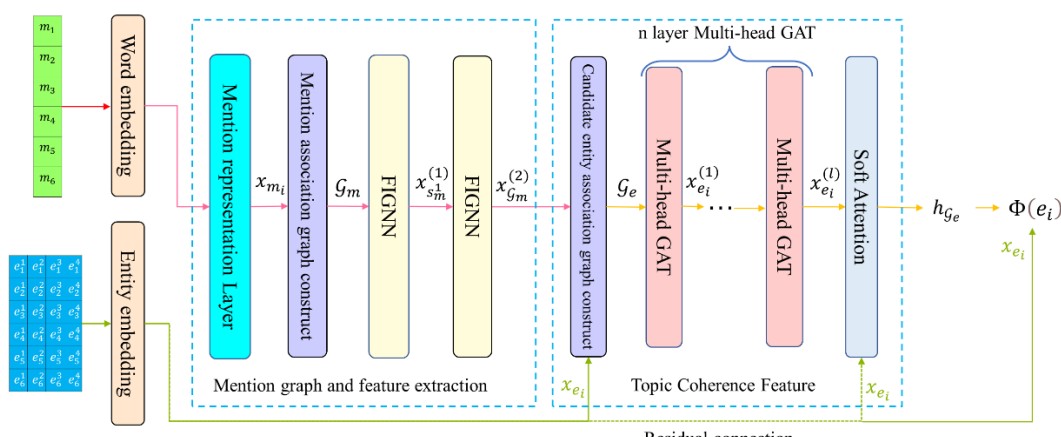

**Figure 1.** The overall architecture of our model. Assume that the current document ($D$) has six mentions, and each mention has four candidate entities.

### 4.1. Mention Association Graph and Feature Extraction

We start by categorizing the possible topics. Select $k$ left neighbor words $c_{i,l} = \{w_{i,l1}, w_{i,l2}, \ldots, w_{i,lk}\}$; $k$ right neighbor words, $c_{i,r} = \{w_{i,r1}, w_{i,r2}, \ldots, w_{i,rk}\}$; and $m_i$ own words, $c_{i,m} = \{w_{i,m1}, w_{i,m2}, \ldots, w_{i,mk}\}$, in the local context. Then, we embed these words into the same vector space. We first sum the word vectors in each of the three parts to obtain the vectors of the three parts: $h_{c_l}, h_{c_m}, h_{c_r} \in \mathbb{R}^d$. Then, we concatenate the three parts, embedding the vectors into one vector ($h_{m_i} \in \mathbb{R}^{d \times 3}$) as the representation vector of $m_i$. First, we use the a two-layer multilayer perceptron (MLP) to reduce the dimensionality of $h_{m_i}$, and then use the self-attention mechanism to obtain the final representation ($x_{m_i} \in \mathbb{R}^d$) of $m_i$:

$$x_{m_i} = \alpha \cdot V \tag{2}$$

$$\alpha = \text{softmax}\left(\frac{QK^T}{\sqrt{d_k}}\right) \tag{3}$$

$$Q = W_Q \cdot h'_{m_i} \tag{4}$$

$$K = W_K \cdot h'_{m_i} \tag{5}$$

$$V = W_V \cdot h'_{m_i} \tag{6}$$

where $W_Q, W_K, W_V$ is the matrix that can be learned; we obtain $h'_{m_i} \in \mathbb{R}^d$ by using a two-layer MLP:

$$h'_{m_i} = \text{Dropout}(W_{linear,1}\sigma(W_{linear,2}(h_{m_i}))) \tag{7}$$

where $W_{linear,1}, W_{linear,2}$ are learnable parameters, and $\sigma$ is the activation function; here, we use Tanh. The model ($x_{m_i}$) is obtained as shown in Figure 2.

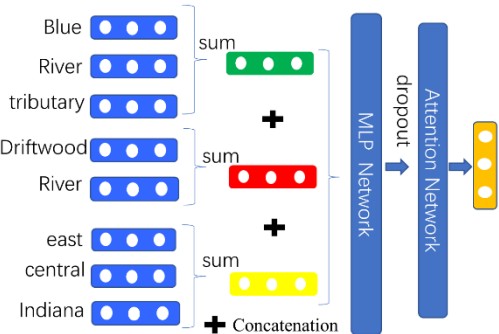

**Figure 2.** The architecture of the mention representation layer.

After obtaining the set of all mention hidden representations ($S_{x_m} = \{x_{m_1}, x_{m_2}, \ldots, x_{m_T}\}$) of the current document ($D$), the full mention connection graph ($\mathcal{G}_m = (\mathcal{V}_m, \mathcal{E}_m)$) is built. We use the mention set ($M$) of the current document ($D$) as the node ($\mathcal{V}_m$) of the graph. The set $\{(m_i, m_j) | m_i, m_j \in M\}$ is obtained by making a Cartesian product of $M$ with itself as a set of edges ($\mathcal{E}_m$). The weight value of each edge in the graph is obtained by calculating the cosine similarity between the two mention representation vectors. The final composition of the full connection mention graph is shown in Figure 3a.

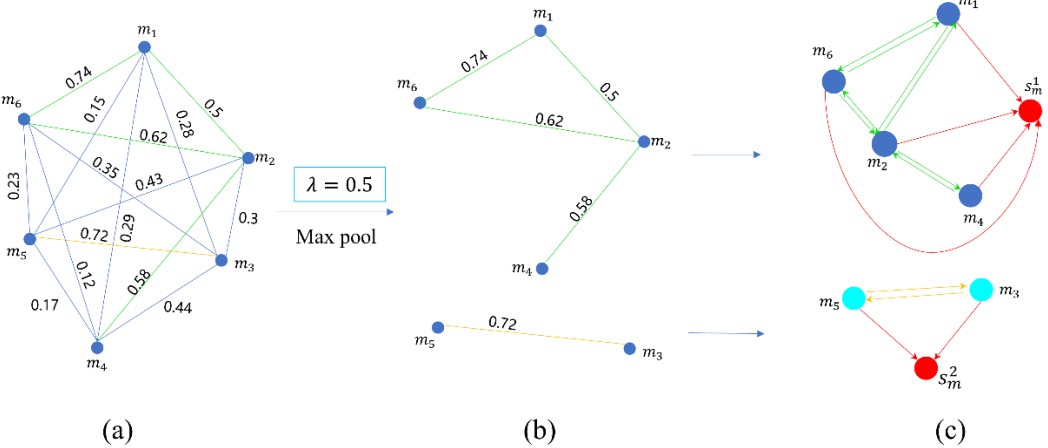

(a)             (b)             (c)

**Figure 3.** The whole process of mention graph construction. We assume that the current document ($D$) has six mentions. (**a**) A fully connected graph of mentions. (**b**) Sparse graph after maximum pooling. (**c**) Mention association graph after adding global nodes and unique edges.

We define a similarity threshold ($\lambda$) to break the edge if the weight value of the edge in the graph is lower than $\lambda$. Finally, multiple sparse mention association graphs are obtained, as shown in Figure 3b. We define the number of obtained sparse graphs as the number of topics in the current document. The whole mention set of document $D$ is split into different $M_1 \cup M_2 \cup \ldots \cup M_n$ according to the situation of sparse connection, where $n$ is the number of mention association graphs after sparse connection (if a node has no connected edges with other nodes, then we treat this node as an entire graph).

Earlier GNN graph readout methods used linear methods, such as summing and averaging of all graph nodes, to obtain a full graph representation. However, this method loses a considerable amount of feature information. Therefore, we propose a new approach to graph readout. We add a global node ($s_m^n$) to each graph. Furthermore, we construct a unique edge between each mention node and the global node; this edge only points from the mention node to the global node, as shown in Figure 3c. Therefore, when we use our variant GNN, the nodes at both ends of this unique edge indicate that the information can only flow from the mention node to the global node. After the multilayer GNN, our global

node can represent the information of the whole graph. The initial representation of the global node is defined as follows:

$$x_{s_m^n}^{(0)} = \frac{1}{T} \sum_{i=1}^{T} x_{m_i}^{(0)} \qquad (8)$$

Existing GGNN networks [16] are mainly applied to graph structures with sequence structures. In order to adapt to our task, we modified the GGNN so it only focuses on in-degree information of the graph neural network (FIGNN). Specifically, we modify the form of the adjacency matrix in the GGNN from the original form considering the splicing of the in-degree adjacency matrix and the out-degree adjacency matrix to a form considering only the in-degree adjacency matrix. Considering the information-capturing capability of GRU, we only use a single-layer linear transformation as the message-passing function and update function in the GNN. $E_{in}(i) = [(j_1, i), (j_2, i), \ldots, (j_{d_i}, i)]$ represents the edge from node $j_{d_i}$ to node $i$, where $j_{d_i}$ is the in degree of node $i$. After the mention association graph is constructed, we use an FIGNN to capture the collective information of the mention association graph. For the representation vector $(x_{m_i}^{(l)})$ of each node in graph $\mathcal{G}_m$, the update rule is defined as follows:

$$x_{m_i}^{(l+1)} = W_{g,1}^{(l)}(x_{m_i}^{(l)} \big|\big| h_k^{(l)}) \qquad (9)$$

where $\left\{ h_k^{(l)} : 0 \leq k \leq d_i \right\}$ is the GRU hidden state, and the initialization status $(h_k^{(0)})$ is set to the zero vector. The GRU is then used to aggregate information between the hidden layer representation and the node representation passed through the message function:

$$h_k^{(l)} = GRU^{(l)}(W_{g,2}^{(l)} x_{m_{j_k}}^{(l)}, h_{k-1}^{(l)}) \qquad (10)$$

After one layer of the FIGNN, we obtain the final representation of all nodes. We use global node representation to represent the whole graph representation:

$$x_{\mathcal{G}_m} = x_{s_m^n}^{(l)} \qquad (11)$$

*4.2. Topic Coherence Feature*

After obtaining the whole graph representation of the mention graph, we move to the next stage, which is extraction of the global coherence feature of the current topic. First, we construct the candidate entity association graph, $\mathcal{G}_e = (\mathcal{V}_e, \mathcal{E}_e)$. We construct a central node. The representation of the central node is the representation of the previously obtained mention graph. Consider all candidate entities in the mention set of the current mention graph as nodes of the graph to obtain the final set of nodes of the candidate entity association graph, $\mathcal{V}_e = \{S_m^n, e_i | e_i \in \Gamma(M_n)\}$. We connect all candidate entity nodes to the central node and obtain the set of edges as $\mathcal{E}_e = \{S_m^n, e_i | e_i \in \Gamma(M_n)\}$. The final candidate entity association graph is shown in Figure 4.

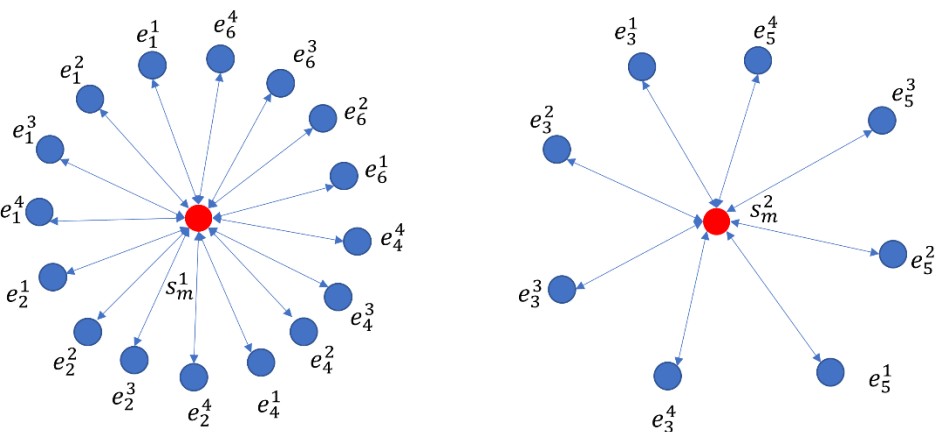

**Figure 4.** The figure shows the candidate entity association graph. The red node is the central node.

After obtaining the candidate entity association graph, we use a multilayer, multi-headed GAT to capture the global coherence feature of the current topic. The representation for each node ($x_{e_i}^{(l+1)}$) in the candidate entity association graph is defined as follows:

$$b_{ij} = a([W_{gat,1}^{(l)} x_{e_i}^{(l)} || W_{gat,2}^{(l)} x_{e_j}^{(l)}]), j \in \mathcal{N}_i \tag{12}$$

where $x_{e_i}$ is the representation vector of candidate entity $e_i$, $x_{e_j}$ is the representation vector of the neighbor nodes of candidate entity $e_i$, $a$ is a single-layer feedforward neural network, $W_{gat,1}^{(l)}$, $W_{gat,2}^{(l)}$ are trainable parameters initialized with different parameters at each layer. Then, we use Softmax to obtain the attention score of each node:

$$\alpha_i = \frac{\exp(\text{LeakyReLU}(b_{ij}))}{\sum_{k \in \mathcal{N}_i} \exp(\text{LeakyReLU}(b_{ik}))} \tag{13}$$

$$x_{e_i}^{(l)} = Max(||_{k=1}^{K} \sigma(\sum_{j \in \mathcal{N}_i} \alpha_{ij} W_{gat,3}^{(l)} x_{e_i}^{(l-1)})) \tag{14}$$

$$x_{e_i}^{(l+1)} = x_{e_i}^{(l)} + x_{e_i}^{(0)} \tag{15}$$

where $K$ is the number of heads of the multihead graph attention network, and the maximum value of each head is the node representation. $x_{e_i}^{(0)}$ represents the initial node representation of the candidate entity association graph. Here, the form of residual joints is used.

To obtain the representation vector of the current graph, we use soft attention to strive for the attention of the central node ($S_m^n$) with other nodes; the final graph level representation is as follows:

$$h_{\mathcal{G}_e} = \sum_{e_i \in \Gamma(M_k)} \beta_i x_{e_i}^{(l+1)} \tag{16}$$

$$\beta_i = \text{softmax}(\varepsilon_i) \tag{17}$$

$$\varepsilon_i = q^T \sigma(W_{linear,3} x_{e_i}^{(l+1)} + W_{linear,4} x_{s_m^n}^{(l+1)} + r) \tag{18}$$

where $q, r \in \mathbb{R}^d$ and $W_{linear,3}$, $W_{linear,4}$ are trainable parameters. Finally, we use the obtained candidate entity association graph representation of the current topic as a global coherence representation of our current topic. We perform a cosine similarity calculation using the current feature with the embedding ($x_{e_i}$) of each candidate entity ($e_i \in \Gamma(M_k)$) in the currently mention set ($M_k$) to obtain the global score of each candidate entity:

$$\Phi(e_i) = \cos \text{inesmliarity}(h_{\mathcal{G}_e}, x_{e_i}) \tag{19}$$

## 5. Rank and Model Training

We focus on the global model, with the local model and the corresponding loss function proposed by Ganea and Hofmann [5]. $\widehat{p}(e|m)$ was also proposed by Ganea and Hofmann [5]. $\widehat{p}(e|m)$ is calculated using two indices; one index is the hyperlink count statistics of mention entities in Wikipedia, and the other index is the hyperlink count statistics of mention entities in a large predictor database; finally, the average probability of these two indices is calculated.

### 5.1. Candidate Entity Ranking

Candidate entity ranking. After obtaining the global score of each candidate entity, we concatenate the local score ($\Psi(m,e)$) and global score ($\Phi(e)$) of each candidate entity with the mention candidate entity prior probabilities ($\widehat{p}(e|m)$) and use a two-layer linear transformation to obtain the score of each candidate entity:

$$\rho_i(e) = W_{linear,5}(\text{Re}lu(W_{linear,6}[\Psi(m,e)||\Phi(e)||\hat{p}(e|m)])) \tag{20}$$

### 5.2. Model Training

Model training. We use max margin loss, and the loss function is calculated as follows:

$$L(\theta) = \sum_{D\in\mathcal{D}} \sum_{m\in D} \sum_{e\in\Gamma(m_i)} h(m_i,e) \tag{21}$$

$$h(m_i,e) = \max(0, \gamma - \rho_i(e^*) + \rho_i(e)) \tag{22}$$

where $e^*$ is the gold entity mention to $m_i$, and $\gamma$ is the hyperparameter. The objective of the global model is to minimize this loss function ($L(\theta)$).

## 6. Experiment

### 6.1. Datasets

We chose the following most popular datasets based on previous research as our experimental datasets. AIDA-CoNLL [19] is an in-domain scenario dataset that contains AIDA-train for the training data of our model, AIDA-A for validation, and AIDA-B for testing. MSNBC, AQUAINT, and ACE2004 are used as out-domain scenario datasets, which were compiled and uploaded by Guo and Barbosa [20]. The data statistics for the datasets are shown in Table 1.

**Table 1.** Datasets statistics.

| Dataset | # Mention | # Doc | Mentions Per Doc | Gold Recall |
|---|---|---|---|---|
| AIDA-train | 18,448 | 946 | 19.5 | - |
| AIDA-A | 4791 | 216 | 22.1 | 97.3 |
| AIDA-B | 4485 | 231 | 19.4 | 98.3 |
| MSNBC | 656 | 20 | 32.8 | 98.5 |
| AQUAINT | 727 | 50 | 14.5 | 94.2 |
| ACE2004 | 11,154 | 36 | 7.1 | 90.6 |

### 6.2. Candidate Entity Selection

For each mention ($m_i$), We use $\widehat{p}(e|m)$ to obtain the 30 candidate entities with the highest scores. Then, we keep the four candidate entities with the highest $\widehat{p}(e|m)$ and keep the four candidate entities with the highest $f(h_e^T, \sum_{w\in n_i} h_w)$ scores, where $h_e, h_w \in \mathbb{R}^d$ are the embeddings of entities and words, and $n_i$ represents the 50 contextual words in the local window of $m_i$. To prove the validity of the model, we use the same pretraining method as in [5] to obtain the embeddings of entities.

### 6.3. Hyperparameter Setting

We adapt our hyperparameters to the model's performance in the AIDA-A validation set. We set the embedding dimension of the model for words and entities to 300. We set each batch to contain only one document. The rank margin is $\gamma = 0.01$. We use Adam [21] as the optimizer and first set the learning rate to $1 \times 10^{-4}$. We set the learning rate to $1 \times 10^{-5}$ when the accuracy of our validation set exceeds 91.5%. Our FIGNN is set to two layers, and the number of layers of the multiheaded graph attention network is set to four. Each layer is initialized with different parameters of the GAT, and the number of heads in each layer is set to six heads. The similarity threshold between mention pairs is set to $\lambda = 0.3$.

### 6.4. Contrast Models

We compare our model with the following existing methods. AIDA [7] designed a global mapping algorithm that consists of two phases, as well as several consistent features, which have some relevance in the domain or category. GLOW [8] uses SVM to combine both local and global features for calculation. RI [22] proposes that there is a relationship between entities and makes an inference. WNED [20] first constructs the disambiguation graph and later uses random walks on this graph. The DEEP-ED [5] model uses LBP for joint inference learning and entity embedding learning at the global level. Ment-Norm [6] models multiple potential relationships that may exist between mentions and, achieving satisfactory results. DGCN [12] proposes a dynamic GCN model that aggregates knowledge from dynamic nodes to dynamically capture global document topic features.

### 6.5. Result

We evaluated our merged model on all public datasets. Because the main focus of the present work is the global model, we focus on comparing methods with consistent local models but inconsistent global models. These models are state-of-the-art methods under consistent local models. We used micro F1 values for this measure.

The results are shown in Tables 2 and 3; our model achieves the most advanced results on both MSNBC and ACE2004 datasets, proving the effectiveness of our global model.

**Table 2.** Micro F1 scores on five out-domain test datasets.

| Method | MSNBC | AQUAINT | ACE2004 |
|:---:|:---:|:---:|:---:|
| Prior $\widehat{p}\left(e\middle|m\right)$ | 89.3 | 83.2 | 84.4 |
| AIDA | 79 | 56 | 80 |
| GLOW | 75 | 83 | 82 |
| RI | 90 | **90** | 86 |
| WNED | 92 | 87 | 88 |
| DEEP-ED | 93.7 | 88.5 | 88.5 |
| Ment-Norm | 93.9 | 88.3 | 89.9 |
| DGCN | 91.0 | 89.4 | 90.6 |
| Our model | **94.9 $\pm$ 0.2** | 88.4 $\pm$ 0.1 | **91.0 $\pm$ 0.1** |

**Table 3.** Comparison of performance scores of global models on the in-domain AIDA-B dataset.

| Method | In-KB acc. (%) |
|:---:|:---:|
| AIDA | 84.8 |
| GLOW | 90.7 |
| RI | 71.51 |
| WNED | 89.0 |
| DEEP-ED | 92.22 |
| Ment-Norm | 93.07 |
| DGCN | 93.13 |
| Our model | **93.2** |

On the in-domain dataset, our model outperforms Ment-Norm, benefiting from the strong learning ability of our global model. For example, our model benefits significantly from the use of the attention mechanism to embed the mention words and entities in the same space.

*6.6. Analysis*

6.6.1. Mention Pair Similarity Threshold Value Learning

The similarity threshold value for mention pairs is crucial for our model. This threshold value determines the number of mention association graphs for each document after max pooling the fully connected mention graph with the number of topics in the same document. The number of mention association graphs in the document is dynamically adjusted according to the learning of the mention embedding vector, which partly weakens the influence of the artificially defined threshold size on the model and enhances the robustness of the model. The impact of the threshold size on the performance of our model is detailed in Figure 5. According to Figure 5, the best results are obtained when the threshold value is 0.3 for both the domain and the cross-domain datasets. The corresponding edge in the fully connected mention association graph is disconnected when the cosine similarity between two entity mention pairs is less than 0.3. Through experimental observation, most of the association graphs mentioned by fully connected entities are divided into two subgraphs, with some small cases of three subgraphs. However, there are few cases with no division. This confirms our suspicion that there are multiple topics in a document.

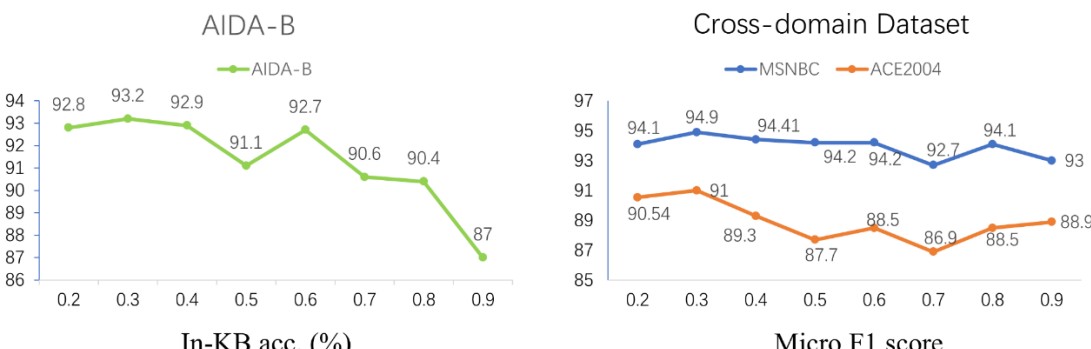

**Figure 5.** Ablation study with varying $\lambda$ values.

6.6.2. The Impact of Residual Networks

The experimental results show that in terms of the representation of the fully connected graph of candidate entities, using GAT with residual connections to represent the fully connected graph of candidate entities can achieve improved results through deeper layers. A multi-head GAT without a residual network (A) and fewer layers can achieve satisfactory results. However, experiments show that using residuals in a deep multihead GAT results in better performance than a shallow GAT without a residual network. We found that in a deep GAT, if the multihead GAT does not use the residual network, the loss is reduced, although performance on the AIDA-A validation set was reduced, demonstrating that the deep multihead GAT overfit the model.

**7. Conclusions**

We hypothesize that a document may have more than one topic. Considering this hypothesis, a global model for multitopic global consistency feature extraction is proposed. We extract each topic's final coherence features through a two-stage graph construction. We creatively propose classifying different topics in the same document by mentioning the similarity. We propose a variant GNN for our model and a new graph readout method. A large number of experiments on public datasets demonstrate the effectiveness of our global model. In future work, we will investigate the optimization of the local model and the optimization of the training speed of the global model.

**Author Contributions:** C.Z.: conceptualization, investigation, methodology, software, validation, visualization, writing—original draft, and writing—review and editing. Z.L.: conceptualization, resources, funding acquisition, and supervision. S.W.: funding acquisition. T.C.: project administration and supervision. X.Z.: software. All authors have read and agreed to the published version of the manuscript.

**Funding:** This work was supported by Qilu University of Technology (Shandong Academy of Sciences) Integration Innovation of Science, Education and Industry Program(2020KJC-ZD16).

**Data Availability Statement:** AIDA dataset is available at http://resources.mpi-inf.mpg.de/yago-naga/aida/download/, accessed on 3 March 2022; MSNBC dataset is available at https://cogcomp.seas.upenn.edu/page/resource_view/4, accessed on 7 March 2022; AQUAINT dataset is available at http://community.nzdl.org/wikification/docs.html, accessed on 3 March 2022 and ACE2004 dataset is available at https://cogcomp.seas.upenn.edu/page/resource_view/4, accessed on 7 March 2022.

**Conflicts of Interest:** The authors declare no conflict of interest.

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
