# Peer review of "Multitopic Coherence Extraction for Global Entity Linking"

_electronics, doi:10.3390/electronics11213638_

Round 1
Reviewer 1 Report
In this paper, the authors construct mention association graphs and candidate entity association graphs to obtain multi-topic coherence features of the same document by graph neural networks (GNN). Overall, the topic of this paper is convincing, and the problem is hot. I hope to see a revised version. And, I have many questions are as follows.
1. Please provide the convergence analysis and complexity analysis of the algorithms.
2. GNN has been used for many years, please give us more explanations about why the model is novel by consulting with the book https://www.deeplearningbook.org/ or “Neural Networks and Learning Machines” by Haykin, S.O.
3. Please edit the English language and check the grammatical mistakes. There are some online tools to do it.
Author Response
I have provided a comprehensive and detailed response to your question in the attached word, please check it out. Thank you.

Reviewer 2 Report
While the topic of discussion is interesting, I am suggesting the following comments to be addressed before this paper can be considered for publication.
Introdcution and literature review do not adequately address the research probelm and Gap. Justification for and the significance for doing this research is not very evident
Results are presented in a good format, however they are not followed by sound discussion.
Author Response

(The authors gave the same response as above.)

Reviewer 3 Report
The paper is well written, there are some minor comments.
Give some more details about equation 1,
Write some more details about Figure 1
Author Response

(The authors gave the same response as above.)

Round 2
Reviewer 1 Report
Please put the convergence analysis and model complexity analysis into the main body of this paper. I can see those things in the response letter but not in the manuscript.
Author Response
The reason I did not include the model complexity and model convergence in response to manuscript is because: The concern in my article is mainly for global consistency of topics, I think there should be more than one topic consistency, not a single topic. For this purpose, I propose a multi-topic consistent feature extraction model. In my experiments I have shown that the whole model converges because I have already obtained the results. The current time complexity is also a result of a simple calculation based on other papers. These two points I think are not very critical for the whole model. The manuscript focuses more on the proposed hypothesis and the use of the model to verify the correctness of this hypothesis. Do I have to include these two points in the manuscript?